# XPA: DNA Repair Protein of Significant Clinical Importance

**DOI:** 10.3390/ijms21062182

**Published:** 2020-03-22

**Authors:** Lucia Borszéková Pulzová, Thomas A. Ward, Miroslav Chovanec

**Affiliations:** 1Department of Genetics, Cancer Research Institute, Biomedical Research Center, Slovak Academy Sciences, 845 05 Bratislava, Slovak Republic; lucia.pulzova@savba.sk; 2Department of Biomedical and Life Sciences, Faculty of Health and Medicine, Lancaster University, Lancaster LA1 4YG, UK; t.ward2@lancaster.ac.uk

**Keywords:** XPA protein, nucleotide excision repair, biomarker, cancer, small molecule inhibitors, single nucleotide polymorphism, prognostic and predictive value

## Abstract

The nucleotide excision repair (NER) pathway is activated in response to a broad spectrum of DNA lesions, including bulky lesions induced by platinum-based chemotherapeutic agents. Expression levels of NER factors and resistance to chemotherapy has been examined with some suggestion that NER plays a role in tumour resistance; however, there is a great degree of variability in these studies. Nevertheless, recent clinical studies have suggested Xeroderma Pigmentosum group A (XPA) protein, a key regulator of the NER pathway that is essential for the repair of DNA damage induced by platinum-based chemotherapeutics, as a potential prognostic and predictive biomarker for response to treatment. XPA functions in damage verification step in NER, as well as a molecular scaffold to assemble other NER core factors around the DNA damage site, mediated by protein–protein interactions. In this review, we focus on the interacting partners and mechanisms of regulation of the XPA protein. We summarize clinical oncology data related to this DNA repair factor, particularly its relationship with treatment outcome, and examine the potential of XPA as a target for small molecule inhibitors.

## 1. Introduction and Nucleotide Excision Repair

Nucleotide excision repair (NER) is a universal and versatile repair pathway capable of removing a broad spectrum of DNA helix-distorting lesions, such as bulky DNA adducts. In addition, it prevents the formation of DNA lesions that act as precursors of DNA-protein adducts [1]. Importantly, it is the sole mechanism in mammals for the repair of two prominent ultraviolet (UV) radiation-induced DNA lesions, cyclobutane pyrimidine dimers (CPDs) and 6-4 photoproducts (6-4 PPs) (reviewed in [2]). Similarly, NER is required for the removal of DNA lesions produced by many chemotherapeutic drugs, such as cisplatin (CDDP), thereby contributing to chemotherapy resistance and clinical treatment outcome [3]. Therefore, dissecting the molecular details of the NER pathway, as well as the structure and regulations of its components, has been investigated in some detail.

The NER pathway consists of more than 30 proteins responsible for DNA damage recognition, verification, incision, excision, gap filling, and ligation. NER can be divided into two sub-pathways, global-genome (GG-NER) and transcription-coupled (TC-NER) NER, differing only in the damage recognition step. In TC-NER, the principal DNA damage sensor is the elongating RNA polymerase II (RNAPII), which becomes blocked at the site of damaged DNA on the actively transcribed strand. Consequently, lesion recognition by TC-NER depends largely on the transcriptional activity of a given gene. Stalled RNAPII recruits the DNA-dependent ATPase Cockayne syndrome group B (CSB) protein. The higher affinity of CSB for RNAPII and its binding to Cockayne syndrome group A (CSA) protein likely helps to backtrack RNAPII allowing the remaining NER factors to access the DNA lesion (reviewed in [4,5]).

In contrast to TC-NER, the initial damage recognition factor in GG-NER is the *Xeroderma Pigmentosum group C* (XPC) protein complexed with the human homologue of yeast Rad23 protein (HR23B). Accordingly, XPC binds to lesions before the other core NER factors [6,7]. It has been hypothesized that XPC-HR23B initially binds to DNA non-specifically and only then searches for the presence of DNA damage, encircling the undamaged DNA strand and sensing single-stranded structures induced by the lesion without interacting with the lesion directly [8]. The kinetic gating model has been adopted to explain how XPC-HR23B finds damaged sites after non-specific binding to DNA. This model suggests that lesion recognition by XPC-HR23B is a result of competition between the residence time of the complex at the lesion and the time required to form the open recognition complex. On damaged DNA, XPC-HR23B resides at the lesion site long enough to form the open complex, while this is not the case on undamaged DNA [9,10]. Another damage sensor in GG-NER is the damaged DNA binding (DDB) complex, consisting of the DDB1 and DDB2 (also known as Xeroderma Pigmentosum group E protein) subunits. DDB is also called UV-damaged DNA-binding (UV-DDB) protein, as it recognizes CPDs and 6-4PPs [11,12,13] and promotes recruitment of the XPC-HR23B complex to these lesions [6,7,14].

To confirm the presence of a DNA lesion, NER employs a second verification step. This step, and all steps acting downstream, are common to both NER sub-pathways. Interplay of transcription factor IIH (TFIIH) and Xeroderma Pigmentosum group A (XPA) protein mediates this step. TFIIH is a large protein complex that consists of 10 different subunits. It is functionally organized into a core and a CDK-activating kinase (CAK) sub-complex. Both the core and the CAK are required for TFIIH to function in transcription initiation, while only the core complex functions in DNA repair. The seven-subunit core contains Xeroderma Pigmentosum group B (XPB) protein, Xeroderma Pigmentosum group D (XPD) protein, p62, p52, p44, p34, and p8. The CAK sub-complex includes the CDK7, Cyclin H, and MAT1 subunits. Three subunits of TFIIH are associated with enzymatic activities: SF2-family DNA-dependent ATPase/helicase activities residing in XPB and XPD, and cyclin-dependent protein kinase activity displayed by CDK7 (reviewed in [15,16]). While the enzymatic function of XPD is dedicated solely to DNA repair [17], XPB activity is required to help promoter opening during transcription initiation [18,19,20]. It is thought that upon ATP hydrolysis, XPB undergoes a large conformational change that has been implicated in stable anchoring to DNA [21,22]. It appears that XPB functions in NER as a double-stranded DNA (dsDNA) translocase that tracks along one of the two DNA strands in the 5′–3′ direction [20], leading to unwinding of the DNA duplex. The resulting single-stranded DNA (ssDNA) segment then serves as an XPD binding site, which may further extend the unwinding and scans the DNA strand to verify the presence of lesions.

TFIIH interacts with XPC-HR23B and loads onto DNA near the lesion via its XPB subunit. Following TFIIH loading, XPA arrives at the lesion [6,23], thereby completing the NER pre-incision complex assembly. XPA interacts both with TFIIH and XPC-HR23B and stabilizes the opened bubble together with the ssDNA binding protein (RPA) [6,24]. A novel role in lesion verification has been suggested for XPA [25] in which XPA assists in the dissociation of CAK from the TFIIH core, which substantially augments its helicase activity and its affinity for ssDNA [26]. Notably, in the presence of XPA, the helicase activity of the TFIIH core is further potentiated, and its blockage by bulky lesions is more pronounced. It has been hypothesized that the TFIIH-XPA interaction likely results in a conformational change in the TFIIH core complex and a transition of TFIIH function from transcription to NER. However, the precise molecular basis of this is not fully understood. Interaction of XPA with some unusual DNA secondary structures configured within the intermediate NER complexes may also play a role [27].

RPA activates the excision repair cross-complementation group 1 (ERCC1)-*Xeroderma Pigmentosum group F* (XPF) and *Xeroderma Pigmentosum group G* (XPG) nucleases that cleave 5′ and 3′ to the lesion, releasing a 24–32 nucleotide fragment containing the lesion [28,29]. The former nuclease is recruited to the lesion by XPA, while the later arrives through its interaction with TFIIH. XPG also replaces XPC in the pre-incision complex [30,31]. The first incision 5′ to the lesion by ERCC1-XPF only takes place after the pre-incision complex assembly is completed, followed by initiation of repair synthesis, 3′ incision by XPG, completion of repair synthesis and ligation of the nick to restore the original DNA sequence. Repair synthesis is mediated by the polymerase activity of the DNA replication machinery and the new DNA fragment is sealed by DNA ligase I or IIIα-XRCC1 (reviewed in [32]). For more comprehensive information on molecular mechanism of both NER pathways, the reader is referred to recent reviews [4,10,27,32,33,34,35,36,37].

## 2. XPA and Its Function in NER

XPA is the key protein in NER important for DNA damage verification and the recruitment of other NER proteins. Human XPA is a 31 kDa, 273 amino acid (aa) protein that is found, primarily, as a homodimer [38]. XPA contains three separated domains (Figure 1A), a central globular domain and dynamically disordered N- and C-terminal domains. The globular core domain possesses a C4 type zinc-finger motif [39] that is essential for the function and stability of XPA. Both poorly structured N- and C-terminal regions provide the XPA protein with flexibility important for its interaction with multiple partners. The N-terminus accommodates a nuclear localization signal (NLS); however, the question is still open as to whether XPA resides in the nucleus, or whether it is normally resident in the cytoplasm and is imported into the nucleus only after DNA damage [40,41,42,43]. Cellular localization of XPA is discussed in more detail in the text below.

The only biochemical activity assigned to XPA is DNA binding. XPA binds both damaged and undamaged DNA strands through the DNA binding domain (residues 98–239) that encompasses a minimal DNA binding domain (MDB; residues 98–219) (Figure 1A), which overlaps the central globular core domain [24,44,45,46]. XPA recognizes sharply bent DNA backbones rather than DNA lesions per se, providing evidence for the requirement of XPA in the DNA damage verification step. Furthermore, XPA can abort DNA incision when the NER complex has assembled erroneously at undamaged sites [47]. XPA binds both the ssDNA-dsDNA (5′ and 3′ flaps) junction and Y junction via direct interaction with the residues K168 and K179 (located within MBD) and K221, K222, K224, and K236 (located within the wider DNA binding domain). These residues, with additional evolutionarily conserved, positively charged residues residing in the DNA binding domain, form a clamp-like DNA binding domain with two linked arms [48]. Whether XPA binds preferentially 5′ or 3′ to the lesion is unclear. In the absence of TFIIH and XPG, XPA has been shown to bind 5′ to DNA-protein photo-crosslinked structures in biochemical assays [49]. However, the reported preference of XPA interacting partners XPC and RPA for DNA with 3′ overhangs suggests XPA localization at the 3′ junction [50]. Importantly, limited proteolysis data using different XPA-DNA junctional complexes suggest that XPA binds dsDNA and ssDNA sequence without preference for either a 3′ or a 5′ ssDNA overhang [51]. In addition, XPA binds double-stranded three-way or four-way Holliday junctions [27,44] and structural intermediates arising during NER [44]. Furthermore, it has been shown that XPA remains in a post-incision complex via interaction with RPA, suggesting a role for this protein in the latter NER stages [49].

XPA functions in conjunction with RPA as the scaffold for the assembly and stabilization of the NER pre-incision complex, organizing the damaged DNA and this complex to ensure lesions are appropriately excised. Interaction of the two proteins is mediated by the 32 kDa subunit of RPA (RPA32) and a motif in the disordered N-terminal domain of XPA [52]. Because RPA32 is tethered to the substrate binding apparatus by a 33 aa flexible linker [53], and RPA32-binding motif on XPA resides in the flexible, disordered N-terminal domain about 50 residues from MBD [52,54], it is still unknown how the activities of these two proteins are coordinated by this interaction. It has been proposed that the critical factor enabling the coordination of XPA and RPA is the direct physical interaction between the MBD of XPA and the 70 kDa subunit of RPA (RPA70), as this interaction is spatially proximate to the binding of both proteins to the NER bubble [55].

Being the key scaffold protein, XPA is destined to be assembled into often reversible and transient complexes to perform its dedicated function. XPA has been shown to interact with proteins involved in every step of NER, from damage recognition to DNA synthesis. In addition, XPA interacts with proteins that function outside of this repair pathway. Understanding the biological function of these interactions depends on the availability of structural information for these complexes.

## 3. XPA Interacting Partners in NER

It has been shown that the XPC-HR23B heterodimer interacts with centrin 2/caltractin 1 (CEN2) via XPC and that, within the heterotrimer, CEN2 and RAD23B cooperate to stabilize and enhance both the specificity and affinity for damaged DNA of XPC [56]. Consequently, CEN2 likely contributes to augmenting rather weak physical interaction between XPC and XPA, thereby facilitating assembly and/or stabilization of the DNA damage recognition complex [57].

Although numerous studies show that XPA has a high affinity for ssDNA-dsDNA junction and other DNA intermediates/structures (see above) [44,46,48,49], it is generally believed that XPA is primarily recruited to damage site via its interaction with TFIIH, which participates in the unwinding of DNA around the lesion to form a DNA bubble [58,59]. The TFIIH interaction site consists of the final 48 residues of the XPA C-terminus [58] (Figure 1B). XPA exhibits lesion-dependent differential effects on TFIIH helicase activity: it enhances translocation of the helicase along undamaged DNA and enhances lesion-induced stalling of this helicase [59]. XPA interacts with TFIIH via the C-terminus of the Trichothiodystrophy group A (TTDA) protein. Consequently, deletion of the first 15 aa of TTDA abolishes XPA binding and strongly decreases the repair function of TFIIH [60].

XPA associates with the RPA32 and RPA70 [52] (Figure 1B) to form the XPA-RPA complex. This complex is an essential component of NER and is generally implicated in damage verification (see above). While several reports have indicated weak selectivity of XPA and RPA (alone or in combination) for damaged DNA [60,61,62,63], it has been shown that the XPA-RPA complex acts as a “double-check” sensor, simultaneously detecting the DNA backbone (recognized by XPA) and base pair distortion (recognized by RPA) [47]. It seems that XPA (in conjunction with RPA) is required to verify the formation and localization of damage specific repair complexes or in control of their three-dimensional assembly [47]. Interestingly, XPA and RPA interact even in the absence of DNA and form XPA_2_-RPA heterotrimeric complexes [38].

Proliferating cell nuclear antigen (PCNA) is an essential protein for DNA replication, DNA repair, cell cycle regulation, chromatin remodelling, and epigenetics. XPA was found to interact directly with PCNA via the AlkB homolog 2 PCNA interacting motif (APIM) [64,65] (Figure 1B). Accordingly, mutating the APIM increases UV sensitivity, reduces repair of CPDs and 6-4 PPs, and induces cell cycle arrest in S phase. It seems that the high affinity XPA-PCNA interaction is fully dependent on post-translational modifications (PTMs) of both partners and is required for the colocalization of XPA with PCNA in replication foci and loading onto newly synthesized DNA [65]. This interaction and colocalization can occur in the absence of DNA damage [65], indicating a role for XPA outside NER.

DDB recognizes a wide spectrum of UV-induced DNA lesions and stimulates their excision [66,67]. XPA interacts with DDB, however the biological significance of this interaction is poorly understood. XPA interacts physically with DDB2 through aa residues 185–226 and this interaction can be seen both in vitro and in vivo [68] (Figure 1B). Mobility shift and DNase I protection assays suggest the formation of ternary complex DDB-RPA-DNA [66]. RPA has previously been shown to interact with ssDNA and XPA and this interaction enhances binding of the NER factors to damaged DNA [66,69]. It is possible that the role for DDB in enhancing XPA recruitment to DNA damage sites is either directly via the interaction of DDB with XPA, or indirectly through efficient RPA-mediated recruitment of XPA onto DNA lesions.

The interaction between XPA and ERCC1-XPF is essential for NER as XPA recruits this nuclease to DNA repair foci [70]. Residues 59–114 and 91–118 of XPA and ERCC1 respectively, are responsible for interaction between these two NER factors [70,71] (Figure 1B). In particular, three highly conserved glycines of XPA, Gly72-74, have been shown to be essential for ERCC1 binding and UV resistance [72]. These residues are found in a short sequence (XPA_67–80_) shown, by a combination of nuclear magnetic resonance (NMR)-derived distance restrains and X-ray crystallography, to complex with the central domain of ERCC1 [72]. NMR and molecular dynamic simulation identified hydrogen bonding between Gly74, Gly73, and Asp70 of XPA; and Ser142, Gln107, and His149 of ERCC1, respectively. The close proximity of DNA binding and ERCC1 interaction domains of XPA raises the question of whether DNA flows from XPA through ERCC1 to the XPF nuclease or whether the ERCC1-XPF encounters the DNA until disassembly of XPA [73]. Notably, the ERCC1-XPA interaction and nuclear localization of this complex is stimulated by transforming growth factor beta (TGFβ) treatment, resulting in more efficient removal of bulky DNA adducts through the induction of NER [74]. Interestingly, CDDP-sensitive gastric cell lines were observed to have insufficient NER activity attributed to impaired nuclear transport of XPA [73].

## 4. Function of XPA Outside NER

Most NER components have additional molecular function to ensure cell viability [31,75]. There is a growing body of evidence to suggest that XPA has several functions outside the DNA damage response (DDR). However, the involvement of XPA in other biological processes remains unclear. Genome wide expression analysis shows that XPA positively influences the expression of a subset of genes important for steroid hormone metabolism and mitophagy [76], consistent with the observation that XPA deficiency leads to mitochondrial dysfunction [77]. These findings may well explain the neurological disorders and sterility common in xeroderma pigmentosum (XP) patients. It is possible that this transcription enhancement by XPA is due to a direct, as yet undefined, role in a transcription factor complex or as a component of an enhancer or mediator complex [76]. However, it is also possible that, in the absence of NER, widespread DNA damage occurs which inhibits transcription [78].

Human population and mouse studies have suggested that XPA is involved in the repair of oxidized DNA bases [79,80]. A preference for oxidative phosphorylation during energy metabolism in neurons and, consequently, increased oxidative DNA damage may partly explain the neurodegeneration in XPA patients [81]. A study associating DNA repair gene polymorphisms with DNA double-strand break (DSB) repair found that levels of phosphorylated histone H2AX 1 h post ionizing radiation was significantly lower in subjects heterozygous for the *XPA* single nucleotide polymorphism (SNP) rs3176683 (for further details on *XPA* SNPs, see text below), suggesting that XPA may also influence DSB repair [82]. 

## 5. XPA Interacting Partners Outside NER

In response to DNA damage, eukaryotic cells arrest cell cycle progression as part of the DDR. However, the exact mechanism of communication between the DNA damage checkpoints and DNA repair pathways remains poorly understood. The cell cycle checkpoint pathway mediated by the ataxia telangiectasia and Rad3-related (ATR) serine/threonine kinase was shown to regulate GG-NER during S phase through direct targeting of XPA [83]. A lack of structural information for ATR kinase, or a model for how it binds to target proteins, has significantly hampered the identification of an ATR-binding motif on the XPA protein. To overcome this limitation, Shell et al. (2009) [84] employed a protein footprinting approach to map the ATR-interaction site. The N-terminal α-helix of the helix-turn-helix motif in the XPA DNA binding site was found to mediate its interaction with ATR (Figure 1C), possibly in concert with ATR interacting protein (ATRIP) [84]. In particular, XPA Lys188 was found to influence the interaction by modulating the stability of the helix [84]. The ATR-XPA interaction physically links DNA damage checkpoints and NER and may represent a novel regulatory mechanism for NER. It is possible that ATR modulates XPA nuclear transport in a cell cycle-dependent manner, as increased cytoplasmic localization in G1 phase and nuclear accumulation in G2 phase has been observed [85,86]. Interestingly, the majority of XPA molecules are localised to the nucleus during G2 phase, independent of DNA damage [85]. Centrosomal protein 164 (CEP164) binds to XPA in a region required for UV resistance (Figure 1C). The XPA-CEP164 interaction is essential for the localization of CEP164 to CPDs and for UV-induced cell cycle checkpoint kinase 1 (CHK1) phosphorylation, further demonstrating a connection between XPA and cell cycle checkpoints. It has been speculated that ATR-mediated phosphorylation of XPA and CEP164 may serve as a signal for DNA repair and the maintenance of activated checkpoints [87]. Members of the nuclear-Dbf2-related (NDR) family of serine/threonine kinases are highly conserved from yeasts to humans and function in processes associated with cell cycle regulation, including centrosome duplication, apoptosis, and the alignment of mitotic chromosomes [88]. Although XPA interacts with NDR1, the biological relevance of this interaction is largely unknown. It appears that NDR1 regulates the removal of XPA from the chromatin without, itself, directly associating with chromatin. Mechanically, this could be achieved by the regulation of the ATR-mediated DDR pathway [89].

XPA also interacts with poly(ADP-ribose) polymerase 1 (PARP1) (Figure 1C), facilitated by the PARylation of XPA. The direct XPA-PARP1 interaction further stimulates PARP1 activity and promotes additional PARylation events and the opening of the chromatin structure. In addition, PARP1 interacts with DDB2 and facilitates DNA damage recognition [90]. Interestingly, PARP1 binding to XPA or DDB2 is sufficient to stimulate PARP1 activity in the absence of DNA strand breaks [91]. PARP1 also forms a stable complex with XPC and rapidly transfers this NER factor to DNA lesions. Based on these multiple interactions with NER factors, it seems that PARP1 has some role in NER, where it may assist in complex formation.

Nitta et al. (2000) [92] employed a yeast two-hybrid screen to conduct an unbiased search for novel binding partners of XPA. Among the positive clones, there were five unknown cDNA. These were designated XPA-binding protein 1-5 (XAB1-5) [92]. XAB1 is a GTP-binding protein, localized mainly in the cytoplasm, with an N-terminal GTP-binding domain required for the GTPase activity. Interestingly, XAB1 binds to the N-terminal region of XPA (Figure 1C), where the nuclear localization signal is located (aa residues 30–42) [93]. It is possible, therefore, that the binding of XAB1 sequesters XPA to the cytoplasm. XAB2 was previously found to interact with RNAPII and two TC-NER specific proteins, CSA and CSB [94]. It has been reported that XAB2 has a function in transcription, pre-mRNA splicing and TC-NER, resulting in embryonic lethality in *xab2* knockout mice [95]. However, the exact role of the XPA-XAB2 interaction in TC-NER remains unclear. XAB3 and XAB5 were identified as a putative metallopeptidase, charged multivesicular body protein 1A (CHMP1A) [96], and a Golgi reassembly stacking protein of 65 kDa (GRASP65) [97], respectively, while XAB4 shares some homology with GRASP65 [92]. The two-hybrid screen also revealed Ras-association domain family 1A (RASSF1A) [98] scaffold protein as a novel XPA interacting partner. This XPA-RASSF1 interaction is essential for XPA to exert its repair activity and promotes its deacetylation. Importantly, a cancer-associated SNP variant, *RASSF1A* A133S, exhibits differential XPA binding, inhibits DNA repair and XPA deacetylation, and hyperstabilizes the XPA-RPA complex. The XPA-RPA complex hyperstabilized by permanent XPA acetylation prevents normal XPA cycling in and out of the nucleus [99].

## 6. Transcriptional Regulation of XPA

The amount of XPA present in any given cell is dynamic and undergoes regulation both at the transcription and post-transcription level, and this can have substantial clinical implications. It is believed that enhanced sensitivity to CDDP observed in some malignancies, particularly in testicular germ cell tumours (TGCTs), results from decreased NER capacity [100] and lower levels of its key factors, such as XPA [101]. Recently, it has been found that CDDP-induced DNA damage formation is uniform and that the accumulation of damage is not driven by damage formation but by the efficiency of repair. In mammals, differences in DNA repair capacity following CDDP treatment are described across different organs and these are associated with tissue-specific transcriptomic and epigenomic profiles [102]. This indicates tissue-specific effectiveness of DNA repair and provides an explanation for differences in CDDP sensitivity/resistance of various tumour types.

Since the circadian clock system is intimately integrated into all metabolic and signalling pathways in a cell, it is not surprising that it also affects the DDR. Recently, it has been reported that the human timeless protein interacts with the ATR-ATRIP complex and CHK1 [103]. Moreover, it was shown that period circadian protein homolog 1 (PER1), a protein important for circadian rhythm in cells, physically interacts with cell cycle checkpoint kinase 2 (CHK2), a serine/threonine kinase that forms part of the ataxia telangiectasia mutated (ATM) protein kinase-mediated cell cycle checkpoint. This interaction is enhanced following DSB formation, directly linking circadian rhythm and DNA repair [104]. Circadian clock further regulates both sensitivity to UV damage and the efficiency of NER by controlling chromatin condensation, mainly through histone acetylation [105]. Furthermore, it has been suggested that circadian oscillations directly influence NER capacity by directly influencing XPA accumulation [81,106]. Indeed, it has been demonstrated that *XPA* transcription is controlled by core circadian clock factors, circadian locomotor output cycles kaput (CLOCK) and brain and muscle ARNT-like protein 1 (BMAL1) (Figure 2A), a transcription factor that binds directly to the promotor region of *XPA* [107]. In a clinical context, it has been observed that the time of administration of DNA damaging chemotherapy influences its toxicity. Identifying the 24 h *XPA* transcript rhythm in human blood samples may therefore help to personalize chemotherapy [107]. It has been shown that in mouse brain, liver, and skin, XPA and NER exhibit robust circadian rhythmicity (reviewed in [108]). In addition, in mouse liver cells, the removal of CDDP-induced DNA damage is strongly dependent on circadian rhythm [109]. The guiding of DNA damaging therapy by the circadian clock represents a novel strategy for maximizing the effectivity of cancer treatment and minimizing adverse side effects.

Hypoxia is often observed in solid tumours as a consequence of excessive proliferation and inadequate oxygen supply. Several NER factors contain multiple hypoxia response elements (HREs) in the promoter region of their genes [110]. Hypoxia-inducible factor 1 alpha (HIF-1α) has been shown to regulate XPC and XPD levels after UV radiation and is involved in removal of 6-4PPs and CPDs [110]. XPA is also a direct HIF-1α target and it has been reported that HIF-1α binding to HRE in the promoter region of *XPA* strongly upregulates *XPA* expression [111] (Figure 2B). It is tempting to speculate that low HIF-1α levels resulting in low XPA levels may lead to reduced repair of CDDP-induced DNA damage in TGCTs, which could explain the innate CDDP susceptibility of this tumour type [112]. In lung cancer (LC) cell lines, in which endogenous XPA levels are higher, inhibition of HIF-1α reduces the expression of *XPA*, while in LC cell lines with lower endogenous XPA, hypoxia elevates expression of *HIF-1α* and *XPA* [111]. Specific inhibition of HIF-1α with opium alkaloid noscapine sensitizes ovarian cancer (OC) cells to CDDP and downregulation of HIF-1α correlates with CDDP-induced apoptosis [113]. Noscapine treatment was also found to inhibit glioma and prostate cancer growth [114,115]. The combination of CDDP with specific inhibitors of HIF-1α may provide an attractive strategy for improvement of cancer therapy outcomes.

The non-histone high-mobility group A (HMGA) proteins possess intrinsic transcriptional activity and act as promoter transactivators through the modification of DNA structure and the recruitment of transcription factors. They bind to nucleosome-free short AT-rich stretches in complex and modulate the DNA binding activity and specificity of the targeted transcription factors [116,117]. It has been shown that cells overexpressing HMGA exhibit increased UV sensitivity and decreased cell viability, a hallmark of NER deficiency [118]. One possible explanation is that NER efficiency is affected by the interaction of this repair pathway with other proteins binding to DNA helix distortions leading to the limited accessibility of early NER repair factors to DNA lesions. Given the role of the HMGA proteins in the regulation of transcription, another possible explanation would be that HMGA mediates the negative regulation of *XPA* expression. Indeed, comparison of transcriptome profiles of cells with different HMGA1 statuses has shown that HMGA1 unexpectedly downregulates the expression of genes involved in DNA damage recognition and repair, including *XPA* [119] (Figure 2C). HMGA1 binds directly to the *XPA* promoter in an A/T-rich negative regulatory region and overexpressing of HMGA1 leads to an overall decrease in XPA levels [118].

Recently, a novel mechanism regulating *XPA* expression has been discovered, mediated by a Ca^2+^-dependent C-type lectin domain family 4 member M (CLEC4M) [120]. This type-II transmembrane protein, consisting of an intra-cellular N-terminal domain, a tandem-repeat neck domain and a C-type lectin carbohydrate recognition domain [121,122] recognizes a range of pathogens and mediates the endocytosis of ligands. Emerging evidence has suggested that CLEC4M has a role in tumour progression and metastasis [123,124,125]. In this context, a role of CLEC4M in *XPA* expression regulation might be of high importance. CLEC4M knockdown inhibits *XPA* expression and leads to increased sensitivity to CDDP, while *CLEC4M* overexpression upregulates *XPA*. Interestingly, while the *XPA* mRNA levels were increased in LC cells, an increase at the protein level was not observed. Hence, it appears that CLEC4M may influence DNA repair by regulating *XPA* expression. In addition, it suggests that a role of XPA in CDDP resistance/sensitivity is interconnected with CLEC4M [120].

## 7. Post-Translational Modifications of XPA

NER proteins frequently undergo PTMs to modify their activity. Phosphorylation, ubiquitination, and acetylation of key NER proteins have been shown to both positively and negatively regulate NER function. Regulation of XPA protein–protein interactions after DNA damage requires phosphorylation by the DNA damage checkpoint kinase ATR, deacetylation by the silent mating type information regulation 2 homologue 1 (SIRT1) and ubiquitination by the HECT and RCC1-like domain containing E3 ubiquitin protein ligase 2 (HERC2) (reviewed in [126]).

XPA is phosphorylated at Ser196, located in DNA binding domain, by ATR in a UV dose-dependent manner. This phosphorylation occurs in response to replication fork stalling in the later stages of lesion removal [83]. Phosphorylation of XPA is required to allow XPA to complex with the RPA subunit RPA70 and a Ser196 substitution decreased the affinity of XPA for RPA70 [127]. Phosphorylated XPA (pXPA) is predominantly chromatin-bound and appears to be essential for optimal cell survival after UV radiation. Following the completion of DNA damage repair, the downregulation of NER activity and removal of DNA-bound multiprotein complexes must take place. This is, in part, achieved by the dephosphorylation of NER factors. The wild-type p53-induced phosphatase 1 (WIP1), for example, has been found to play a key role in the downregulation of DNA repair (reviewed in [128]), and has been shown to catalyse dephosphorylation of XPA on Ser196, reducing NER activity [129]. In vivo experiments with *wip1* knockout mice showed that CPDs were repaired more quickly than in wild-type mice, resulting in lower level of UV-induced apoptosis. The observed negative effect of WIP1 on repair activity suggests its oncogenic potential [129].

The NAD^+^-dependent deacetylase SIRT1 regulates various cellular processes including cell metabolism, survival, and the stress response. The exact role of SIRT1 in the regulation of DNA repair activity is still ambiguous. It has been shown that the loss of the *SIRT1* expression significantly reduces the repair rate of CPDs and 6-4PPs in UV-irradiated cells and is associated with downregulation of XPC [130]. While only a small fraction of XPA is subject to acetylation, as observed in mouse liver [131], it has been reported that acetylation on Lys63 and Lys67 reduces XPA activity by interfering with the XPA-RPA interaction and, possibly, interactions with other NER factors [132]. SIRT1 deacetylates XPA at residues Lys63, Lys67, and Lys215 to promote the interaction of ATR with XPA, and the subsequent phosphorylation of XPA Ser196 [133]. Accordingly, it has been shown that increased levels of SIRT1 result in an increase in deacetylated XPA that persists in chromatin-bound state and facilitates repair of UV- and CDDP-induced DNA lesions [134]. Expectedly, increased levels of SIRT1 in cancer cells can confer CDDP resistance and thus represent treatment obstacle [132,134] that is overcome by targeting by SIRT1 inhibitors [135,136,137].

HERC2 ubiquitinates XPA leading to its degradation by the ubiquitin-proteasome complex. HERC2-mediated regulation of XPA contributes to the short half-life and daily oscillation of XPA [107]. DNA damage inhibits XPA proteolysis by promoting the dissociation of HERC2 E3 ligase from XPA and supporting a tight association between XPA and chromatin [131]. It has been demonstrated that XPA phosphorylation on Ser196 enhances the XPA protein level by inhibiting HERC2-mediated ubiquitination upon UV exposure, indicating an antagonizing effect of ATR-mediated phosphorylation on HERC2-mediated XPA degradation [127]. It was observed that downregulation of the HERC2 E3 ligase results in ∼2-fold increase in XPA protein and causes a proportional increase in the rate of repair of both CPDs and 6-4 PPs [131].

Poly(ADP-ribosyl)ation (PARylation) is a reversible PTM that influences enzymatic activity, spatio-temporal localization, protein–protein interactions, and protein turnover, particularly of proteins required for the DDR, including NER [138,139]. XPA has been shown to be PARylated rapidly following UV-radiation, facilitating its recruitment to the site of DNA damage [91] and promoting its interaction with PARP1, with the latter being essential for proficient NER [140]. XPA binds PAR polymerase with high affinity, favouring binding of long PAR chains (55-mer) over short ones (16-mer) [99,141]. The PAR binding site of XPA is located at the interface of the N-terminal DNA binding and DDB2 binding domains, as well as the C-terminal TFIIH binding domain [58,68,142]. Importantly, XPA itself strongly stimulates PARP1 enzymatic activity, clearly indicating that XPA and PARP1 regulate each other in a reciprocal and PAR-dependent manner, potentially acting as a fine-tuning mechanism for the spatio-temporal regulation of the two factors during NER [99]. Interestingly, SIRT1 and PARP1 physically interact, suggesting these reciprocal regulations of XPA are linked [143].

The data described above suggest a network of inter-linked PTMs, that influence turnover, localization and activity of XPA that regulates NER. The exact molecular basis and dynamics of the XPA-ATR-HERC2, XPA-ATR-SIRT1, and XPA-PARP1-SIRT1 network is still unknown and require further study. The functional crosstalk between PTMs taking place to control the steady-state level of XPA may present a novel control mechanism of NER pathway (Figure 3).

## 8. XPA Inhibitors and Their Potential in Combination Cancer Therapy

The fact that XPA patients exhibit the most severe phenotype among all XP patients highlights a critical, and perhaps exceptional, cellular role for XPA. In addition, XPA is unique among NER proteins as it is required for both TC-NER and GG-NER and is essential for the removal of all DNA lesions repaired by NER, including those induced by many chemotherapy regimens. XPA is, therefore, an attractive candidate for targeted cancer therapy and the development of small molecules capable of blocking XPA function is a highly exciting field of translational research. Such molecules in combination with CDDP may represent a novel strategy to improve treatment outcomes, particularly in CDDP-resistant tumours.

Potential XPA inhibitors were initially identified in screens for novel NER inhibitors and small molecules that potentiate CDDP lethality [144,145,146]. The molecules only partially suppressed NER function and were largely non-specific. To improve efficacy and specificity, in silico screening of a virtual small molecule library was used, identifying three candidate molecules, X57, X60, and X80, capable of inhibiting the interactions between XPA and various DNA substrates [147]. Of the three molecules, X80 showed the greatest activity, inhibiting up to 95% of the interactions between XPA and either ssDNA or dsDNA, with or without a CDDP lesion [147,148]. This molecule inhibited all XPA interactions with all substrates equally, suggesting a single mechanism for DNA binding. Molecular modelling and docking analysis of X80 with XPA suggested that a benzoic acid moiety of compound X80 interacts with XPA Lys137 salt bridge, and this is a critical determinant of inhibitory activity [147]. To identify inhibitors with improved potency, two commercial libraries X80 analogues with 85–95% structural similarity were searched. Approximately 30 commercially available analogues were identified and tested for their ability to inhibit XPA–DNA interaction, with compounds able to inhibit greater than 80% of interactions taken forward for further study [149]. This approach further validated the molecular docking data and indicated that improved inhibitory activity of X80 analogues is strongly related to improved hydrophobic interactions in the binding pocket of XPA [149]. These novel XPA inhibitors are promising compounds for the development of anticancer drugs to be used in combination therapy.

The XPA–ERCC1 interaction is essential for a proficient response to CDDP (reviewed in [150]). The availability of a crystal structure for XPA–ERCC1 [72] has allowed for the rational design of inhibitors of this interaction. The employment of a sophisticated relaxed complex scheme docking approach led to the identification of AB-00026258 (also known as NER inhibitor 01; NERI01), a novel selective inhibitor of the XPA-ERCC1 interaction [151]. NERI01 activity was validated by sensitizing lung and colon cancer cells to UV radiation and CDDP in vitro. Docking simulations revealed that the binding between ERCC1 and XPA is primarily mediated by five residues on XPA (Gly72, Gly73, Gly74, Phe75, and Ile76) and 10 residues on ERCC1 (Arg106, Gln107, Gly109, Asn110, Pro111, Phe140, Leu141, Ser142, Tyr145, and Tyr152). NERI01 binding is mediated by a hydrogen bond network within the binding site, making six hydrogen bonds with ERCC1 and stabilizing the interaction between the side chains of Phe140 and Asn110. This creates a hydrophobic cleft for the aromatic regions of NERI01 [121]. Further screens of structurally similar molecules identified AB-00027849 and AB-00026258 as more potent inhibitors of the XPA-ERCC1 interaction [152]. More studies are required to validate these compounds as potential sensitizers of CDDP resistance and to examine any clinical relevance. For detailed information on structural formulas of all mentioned XPA inhibitors, the reader is referred to the original papers [147,148,149,151,152].

## 9. XPA Polymorphisms and Cancer Incidence and Treatment Outcome

SNPs are the most common type of germline genetic variation, and, with the completion of the HapMap project [153], millions of SNPs are now annotated. SNPs may alter numerous cellular functions, including DDR, through regulation of transcription or protein expression of the related DDR factors, thereby playing critical roles in altering an individual’s susceptibility to cancer risk. Several *XPA* SNPs and their association with cancer risk have been studied; however, these data have often been unclear and inconclusive (see text below) very likely due to different roles for XPA in different cell types or tissues. Moreover, a linkage disequilibrium between *XPA* SNPs and other SNPs located close to *XPA* might be a possible explanation for this inconsistency [154]. Some *XPA* SNPs are described in Table 1 and in greater detail below.

The *XPA* rs2808668 and rs10817938 SNPs cause T to C transitions in the 5′-untranslated region (UTR) in the nucleotides -2718 and -514 from the transcriptional start site, respectively. These SNPs are located at transcription factor binding sites, impacting *XPA* mRNA level, which may influence the cellular response to platinum-based chemotherapies. Indeed, rs10817938 heterozygous CT and homozygous TT genotypes have been associated with longer overall survival (OS) in colorectal cancer (CRC) patients receiving oxaliplatin-based chemotherapy [155]. In addition, these SNPs have been shown to be associated with the risk and development of numerous cancers and display significant gene-environment interactions (see below). Recently, the most comprehensive meta-analysis for rs10817938 and rs2808668 and cancer risk (33 types of cancer were examined) showed that harbouring rs10817938 homozygous CC genotype, C allele, and CC/CT genotype in a dominant setting associates with an increased overall cancer risk, with a specific association with digestive system cancers. In contrast, there was no association with overall cancer risk for rs2808668, albeit subgroup analysis revealed a decreased risk in the majority of cancers examined, with the exception of digestive system cancer [156]. In addition, this SNP interacts with environmental factors, such as alcohol consumption in gastric cancer (GC), and clinical–pathological characteristics, such as tumour size, metastatic status at onset, and mitotic index in gastrointestinal stromal cancer [157,158]. These findings highlight an obvious divergence between the two SNPs with respect to cancer risk, although they both reside in the promoter region of *XPA*. This might partially be explained by the environmental factors or clinical–pathological parameters that interact with the *XPA* genetic variants synergistically, contributing to the process of carcinogenesis.

Meta-analysis examining rs1800975 and the risk of developing breast cancer (BC) suggests a decreased risk of developing this malignity in non-Asian populations in a recessive setting [159]. Individuals carrying the TC and CC genotypes at rs10817938 had significantly greater risk of developing oral squamous cell carcinoma (OSCC) compared to the TT genotype. Moreover, OSCC patients with the C allele at this SNP were more susceptible to lymph metastases, poor pathological differentiation and late tumour node metastasis (TNM) stage. This indicates that rs10817938 is a useful biomarker for poor prognosis in OSCC patients. Importantly, a significant gene-environment interaction between smoking and the CC genotype was observed. At the molecular level, T to C substitution at rs10817938 significantly decreased transcription of the *XPA* gene, and therefore the XPA mRNA and protein levels were accordingly decreased. In contrast with rs10817938, no significant association of rs2808668 with OSCC risk or prognosis was observed [160]. The *XPA* rs10817938 SNP was also associated with hepatocellular cancer (HCC) risk in stage 1, where the CC genotype displays an increased risk compared with the TT wild-type and TT plus TC genotype [161]. It also contributes to an increased CRC risk in its variant homozygote and recessive model both in overall and stratification analyses [162].

Another SNP present in 5′-UTR of *XPA*, rs1800975 (A23G), is A to G transition in the nucleotide -4 from ATG start codon having an implication for the binding of the 40S ribosomal subunit and, consequently, the level of XPA protein in the cell. Notably, this SNP was shown to affect DNA repair capacity: one or two copies of the wild-type allele results in significantly higher DNA repair capacity in a host cell reactivation assay [163] and human population studies in healthy cancer-free individuals [164]. Moreover, LC patients with this SNP show an increased response to platinum-based therapy [165]. A23G has been shown to contribute to the risk of developing LC [166,167,168], basal and squamous cell carcinoma (SCC) [169], esophageal SCC (ESCC) [170,171], OSCC [154,172], and OC [173], but not testicular [174], prostate [175,176], colorectal [177], and gastric cancers [178], SCC of the oropharynx [179], or melanoma [180]. The risk associated with A23G and LC is greatly associated with environmental factors. While the G allele is associated with a worse outcome in non-smoking individuals and the young [163,164,181], the A allele is associated with poor outcome in heavy smokers [166,167], with these individuals 3-fold more likely to develop LC [182]. The A23G SNP was also included in a screen for genetic factors predisposing to *TP53* mutations in LC patients and was found to be significantly associated with the prevalence of mutations in this gene, suggesting that it may modulate the occurrence of the *TP53* mutations, thereby contributing to LC [183]. A23G has been associated with chemoresistance in acute myeloblastic leukaemia (AML), where individuals with the AA genotype had a probability of resistant disease 2- and 5-times lower than those with heterozygous AG and homozygous GG variant genotype, respectively. In the multivariate model, the GG genotype was the only independent factor for increased risk of resistant disease. Twice as many AML patients with the GG genotype were either chemoresistant or died during the induction, when compared to the other A23G genotypes [184]. This SNP also plays an important role in response to radiotherapy in head and neck SCC (HNSCC) [185].

G709A SNP (rs number not available) resides in the protein coding region of *XPA* and leads to a G to A transition in exon 6 [186], and thus Arg to Gln substitution at position 228 in the protein sequence. In contrast to A23G SNP, the G709A SNP has virtually no impact on the repair of UV- and benzo(a)pyrene diol epoxide-induced DNA damage compared to wild-type *XPA* [186,187], and appears to have a protective effect for LC patients [166,167,168].

Three further SNPs reside in intron sequences of *XPA*: rs3176658, which causes C to T transition, rs3176721, a C to A transversion, and rs2808667, a T to C transition (www.ncbi.nlm.nih.gov/snp). rs3176658 and rs3176721 are associated with efficacy of platinum-based chemotherapy in LC [188], while rs3176658 alone has been shown to be significantly associated with LC risk [189] and with response to neoadjuvant radiochemotherapy treatment of locally advanced rectal cancer (RC) [190].

## 10. *XPA* Expression as a Cancer Risk Factor and Its Prognostic and Predictive Value

Changes in the expression level of *XPA* are assumed to significantly contribute to cancer risk, disease prognosis and treatment outcome. To identify the potential role of *XPA* mRNA expression in different cancer types, Wu et al. [156] used two approaches: (i) comparison of the *XPA* mRNA expression in 13 types of cancers from the RNA sequencing dataset platform GENT (Gene Expression Database of Normal and Tumor Tissues, http://medical-genome.kribb.re.kr/GENT/) and (ii) freely available information on *XPA* expression in 19 types TCGA (The Cancer Genome Atlas) datasets and their normal controls from ONCOMINE, a cancer microarray database and web-based data-mining platform (https://www.oncomine.org/). These approaches revealed decreased expression of *XPA* in a range of tumours, including bladder, kidney, liver, lung (also seen in [195]), prostate, and stomach cancers. To explore the impact of *XPA* mRNA expression on clinical outcome, Wu et al. [156] examined GEO and Array Express datasets in the Prediction of Clinical Outcomes from Genomic Profiles (PRECOG, http://precog.stanford.edu) database, identifying that higher *XPA* expression is associated with improved OS and a better prognosis in glioma and BC [156]. Furthermore, XPA protein expression has been reported to be significantly decreased in CRC tissues, and patients with high XPA protein expression had longer OS [196,197], though contradictory data also exist [198]. Stratified analysis suggests that this improved prognosis for high *XPA* expressing tumours is particularly relevant for patients who are over 60 years with RC, without distant metastasis, without tumour deposits, and with a tumour diameter >4 cm. These data suggest that XPA might serve as predictive biomarker for prognosis in CRC patients [196]. These studies examined XPA protein levels in CRC patients: further analysis of *XPA* mRNA in CRC compared with matched normal tissues using the ONCOMINE database suggested no change in *XPA* expression [199]. This indicates that protein rather than mRNA expression level is of more clinical relevance in CRC and furthermore, it illustrates that the posttranscriptional/posttranslational regulation might play a more important role in determining XPA protein level than mRNA expression per se.

Statistically significantly lower *XPA* expression level has been found in HNSCC patients compared with controls [200]. In this malignity, no correlation between *XPA* expression and OS was found when looking at the overall HNSCC patient cohort. However, subsite analysis revealed that high *XPA* expression showed a significantly increased OS in patients with SCC of the oropharynx, indicating that it may function as a predictive marker for increased OS in these patients [201].

A possible association between *XPA* expression in tumour tissue and the efficacy of neoadjuvant chemotherapy (NAC) was investigated for locally advanced uterine cervical cancer (UCC). It was shown that the patients who responded to NAC displayed significantly lower *XPA* expression than those with ineffective NAC response, indicating that low *XPA* expression may be a predictive biomarker of NAC efficacy for patients with locally advanced UCC, which may be helpful for improving their prognosis [202].

The *XPA* expression was also detected in cancer tissues from locally advanced nasopharyngeal carcinoma (NPC) patients treated with platinum-based chemoradiotherapy to examine if it is a prognostic factor. The data showed that even though it was not associated with clinical–pathological characteristics, it can act as a prognostic factor for OS and PFS: high *XPA* levels predicted a poor prognosis. In addition, the *XPA* expression together with T and N classifications were independent prognostic factors that can successfully be used for classification of NPC patients into low, medium, and high risk groups for platinum-based chemoradiotherapy, suggesting that XPA levels may be a potential predictor of prognosis in these patients treated with platinum-based chemoradiotherapy, and helpful for selecting patients likely to need and benefit from this treatment in future [59].

In OC, effusion specimens from patients who had a complete response to chemotherapy expressed significantly higher levels of the XPA protein than those who had a partial or no response, arguing against a significant role for XPA in mediating cellular resistance to CDDP in OC. Regarding disease progression, the XPA protein expression in primary diagnosis effusion specimens showed no correlation with PFS or OS. However, in ≥25% of tumour cells in specimens from patients with disease recurrence it predicted better PFS. Similarly, an improved OS was demonstrated in effusion specimens from patients presenting with first disease recurrence. In multivariate analysis of PFS, the XPA protein level was an independent predictor of better outcome [203]. Correlation with residual disease volume was found for *XPA* expression in advanced-stage serous OC [204], although in this study *XPA* expression was unrelated to survival. This discrepancy likely reflects different methodology, as well as the fact that protein and mRNA expression do not fully overlap in many cases.

To evaluate the role of XPA in the response of TGCTs to CDDP-based chemotherapy, Mendoza et al. [174] showed no difference in the *XPA* expression level between non-seminomatous TGCT patients sensitive to CDDP and those not sensitive to this drug. Recently, we have brought evidence that TGCT patients with low *XPA* expression have significantly better OS than patients with high expression. In addition, we have demonstrated that *XPA* expression was increased in the non-seminomatous histological subtype, poor prognosis group according to International Germ Cell Cancer Collaborative Group (IGCCCG), increasing S stage, as well as the presence of lung, liver, and non-pulmonary visceral metastases [205]. In addition, XPA was identified as independent risk factor of poor OS in HCC [206].

## 11. Conclusions

XPA is a component of the pre-incision complex with a role in sensing/verifying DNA damage, recruiting other repair factors and stabilization of NER intermediates. However, the recent studies suggest that the main function of XPA in the cell could have initially been misinterpreted. High affinity of XPA to DNA junction complexes formed during other DNA metabolic processes (e.g., replication, recombination), identification of numerous interacting partners and involvement in Hutchinson–Gilford progeria syndrome [207] indicate important non-NER biological functions of XPA.

An effective DNA repair, allowing cancer cells to survive, grow and proliferate, is often the basis of cancer therapy failure and recurrence of disease. Considering the importance of XPA for both NER sub-pathways, inhibition of the scaffolding activity of XPA by small molecules, their combination with DNA damaging anticancer agents, controlling the subcellular level of XPA, its cytosolic-to-nuclear translocation and stabilization, and finely tuned regulation of XPA PTMs and their crosstalk, may all represent novel promising approaches for the improvement of cancer treatment outcome. In a personal context, *XPA* genetic variations and expression level might once be screened for predicting cancer prognosis leading to additional improvement and a precise approach in cancer treatment.

## Figures and Tables

**Figure 1 ijms-21-02182-f001:**
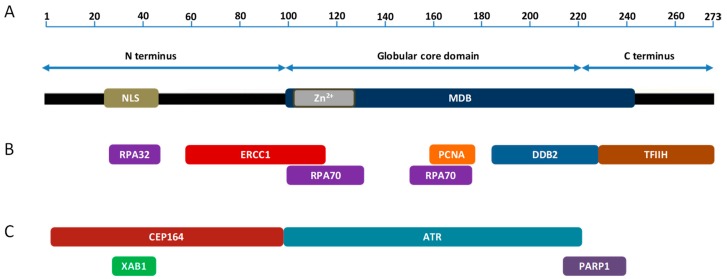
XPA domains and interacting partners. Domain structure of the human XPA protein (**A**). XPA interaction partners involved in NER (**B**). XPA interaction partners outside NER (**C**). Only those partners are shown for which the binding sites on XPA have been mapped. Not to scale.

**Figure 2 ijms-21-02182-f002:**
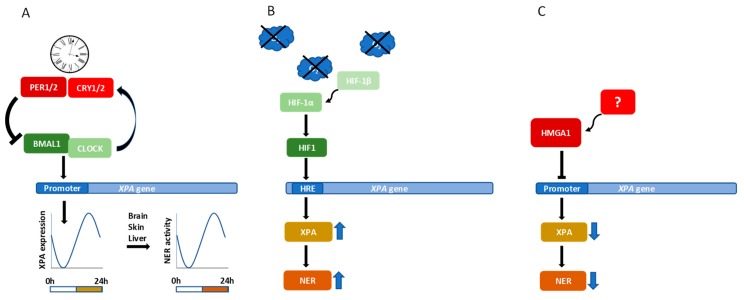
Transcriptional control of the XPA protein level. Transcription factors BMAL1 and CLOCK heterodimerize and drive the transcription of their negative regulators PER and CRY. This negative feedback loop sets up the rhythmic oscillation and drives circadian clocks. The BMAL1/CLOCK heterodimer also regulates the expression of *XPA* resulting in rhythmic oscillation of the XPA intracellular level and NER efficiency (**A**). In hypoxia, HIF-1α forms a dimer with HIF-1β. After translocation to nucleus, the HIF-1 heterodimer binds the HRE in promoter region of the *XPA* gene and upregulates expression of XPA leading to an increased NER efficiency (**B**). The HMGA1 protein binds to negative regulatory element in promoter region of the *XPA* gene and represses its transcription (**C**).

**Figure 3 ijms-21-02182-f003:**
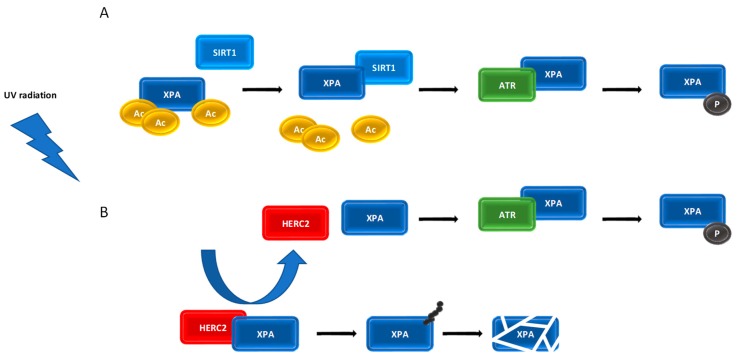
Proposed ATR-SIRT1-XPA and ATR-HERC2-XPA axis. UV-induced DNA damage activates the ATR kinase, which promotes SIRT1 localization at the damage sites and SIRT1-mediated deacetylation of XPA. Deacetylated XPA is a substrate for ATR-mediated phosphorylation. Phosphorylated XPA enhances the repair of damaged DNA (**A**). Upon UV radiation, ATR facilitates the dissociation of the HERC2-XPA complex and prevents XPA ubiquitination and subsequent degradation (**B**).

**Table 1 ijms-21-02182-t001:** *XPA* SNPs and their cancer relevance.

SNP ID	Location	Allelic Variant	Effect	Association with Cancer Risk	Response to Therapy	Reference
rs2808668	5′-UTR	T/C	Binding of transcription factors	No association with cancer risk within overall analysis; Decreased cancer risk with the exception of digestive system cancer in subgroup analysis; No association with OSCC risk and/or prognosis	NA	[156,160]
rs10817938	5′-UTR	T/C	Binding of transcription factors; Decreased transcription of the XPA gene	Homozygous CC genotype, C allele, and CC/CT genotype in dominant setting associates with an increased cancer risk within overall analysis; TC and CC genotypes display higher risk of developing OSCC compared to the TT genotype; It associates with HCC risk in stage 1, where the CC genotype displays an increased risk of HCC compared with the TT wild-type and TT plus TC genotype; It contributes to an increased CRC risk in its variant homozygote and recessive model both in overall and stratification analyses	CT and TT genotypes have longer OS in CRC patients receiving oxaliplatin-based chemotherapy	[155,156,160,161,162]
rs1800975	5′-UTR	A/G	Binding of 40S ribosomal subunit	No association with BC risk in the pooled analysis for all genetic settings; In subgroup analysis, it decreases BC risk in some ethnic groups; GG genotype shows an increased LC risk in some ethnic groups; When combined with rs3176752, it increases neuroblastoma risk; It contributes to a risk from basal and SCC, oral SCC, and OC; AG and GG genotypes significantly decrease the ESCC risk compared to AA genotype; No association with risk of testicular, prostate, and gastric cancers, CRC, SCC of the oropharynx, and melanoma	No association with chemotherapy efficacy and prognosis in EC; Homozygous GG genotype shows a higher response rate than the GA or AA genotype in LC; The GA and AA genotype has an increased risk of death in inoperable LC treated with radiotherapy with or without platinum-based chemotherapy; It plays an important role in response to radiotherapy in HNSCC; The AG genotype imposes with a higher risk of mortality after cancer treatment compared with the GG genotype; No association with OS or disease progression regarding clinical outcome to 5-fluorouracil/oxaliplatin combination therapy in refractory CRC	[154,155,159,166,167,168,169,170,171,172,173,174,175,176,177,178,179,180,185,191,192,193]
rs3176658	Intron	C/T	-	Modest association with LC risk	Significantly associates with PFS in LC; Significantly associates with the response to neoadjuvant radiochemotherapy treatment of locally advanced rectal cancer	[188,189,190]
rs3176721	Intron	C/A	-	NA	Significantly associates with toxicity and efficiency of platinum-based chemotherapy in LC	[188]
rs2808667	Intron	T/C	-	Association with risk of EC	NA	[194]
-	Intron	G709A	-	A significant protective effect in AG heterozygotes in LC		[165,167]
rs3176752	3′-UTR	G/T	Binding of microRNA	When combined with rs1800975, it increases neuroblastoma risk	NA	[191]

BC, breast cancer; CRC, colorectal cancer; EC, endometrial cancer; ESSC, esophageal squamous cell carcinoma; HCC, hepatocellular carcinoma; HNSCC, head and neck squamous cell carcinoma; LC, lung cancer; NA, not analysed; OS, overall survival; OSCC, oral squamous cell carcinoma; PFS, progression-free survival; SCC, squamous cell carcinoma; UTR, untranslated region.

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
