# Peer review of "XPA: DNA Repair Protein of Significant Clinical Importance"

_ijms, 2020, doi:10.3390/ijms21062182_

Round 1

Reviewer 1 Report

 As stated in its abstract, the strongest parts of this article are reviews of transcriptional regulation of XPA and the identification partners of XPA, but the introduction of NER and XPA polymorphisms need to be revised before this paper can be considered for publication. 

  1. Why the authors made a comment that the damage recognition step of GG-NER is poorly understood? and there exists an alternative RPA-XPA damage recognition model? The authors should incorporate more recent articles reporting DNA damage recognition and verification of GG_NER such as Mu et al. DNA Repair 71:33-42 and Sugasawa, K. (2019) The Enzymes 45: 99-138 to build a good solid introduction without misleading information. Information reported in references 6 and 7 may be too old for writing an update review of GG-NER damage recognition. 
  2.  The section reviewing XPA polymorphisms and cancer incidence and treatment outcome can be presented without Tab.1, and the length of this section can be shortened by displaying the most significant part of it.
  3. Fig.1 clearly shows how XPA expression is regulated by different physiological conditions, but !A and 1B should be reversed according to the text.
  4. How XPA functions as a scaffold protein in NER should be expanded in the begining of section 2 and sentences regarding interacting amino acid residues in XPA or other factors should be kept to a minimum to avoid boring the readers. 

Author Response

As stated in its abstract, the strongest parts of this article are reviews of transcriptional regulation of XPA and the identification partners of XPA, but the introduction of NER and XPA polymorphisms need to be revised before this paper can be considered for publication. 

  1. Why the authors made a comment that the damage recognition step of GG-NER is poorly understood? and there exists an alternative RPA-XPA damage recognition model? The authors should incorporate more recent articles reporting DNA damage recognition and verification of GG_NER such as Mu et al. DNA Repair 71:33-42 and Sugasawa, K. (2019) The Enzymes 45: 99-138 to build a good solid introduction without misleading information. Information reported in references 6 and 7 may be too old for writing an update review of GG-NER damage recognition.

We have substantially changed part of our manuscript related to GG-NER. In addition, old references have been replaced with the recent ones, as requested. Modified text reads:

In contrast to TC-NER, the initial damage recognition factor in GG-NER is the Xeroderma Pigmentosum group C (XPC) protein complexed with the human homologue of yeast Rad23 protein (HR23B). Accordingly, XPC-HR23B binds to lesions before the other core NER factors [6, 7]. It has been hypothesized that XPC initially binds to DNA non-specifically and only then searches for the presence of DNA damage, encircling the undamaged DNA strand and sensing the single-stranded structure(s) induced by the lesion without interacting with the lesion directly [8]. The kinetic gating model has been adopted to explain how XPC-HR23B finds damaged sites after non-specific binding to DNA. This model suggests that lesion recognition by XPC-HR23B is a result of competition between the residence time of the complex at the lesion site and the time required to form the open recognition complex. In damaged DNA, XPC-HR23B resides at the lesion site long enough to form the open complex, while this is not the case in undamaged DNA [9, 10]. Another damage sensor in GG-NER is the damaged DNA binding (DDB) protein, consisting of the DDB1 and DDB2 (also known as Xeroderma pigmentosum E protein) subunits. DDB is also called UV-damaged DNA-binding (UV-DDB) protein, as it recognizes CPDs and 6-4PPs [11-13] and promotes recruitment of the XPC-HR23B complex to these lesions [6, 7, 14].

To ensure a presence of the lesion, NER employs a second verification step. This step and all steps acting downstream are common to both NER sub-pathways. Interplay of transcription factor IIH (TFIIH) and Xeroderma Pigmentosum group A (XPA) protein mediates this step. TFIIH is a large protein complex that consists of 10 different subunits. It is functionally organized into a core and a CDK-activating kinase (CAK) sub-complex. Both the core and the CAK are required for TFIIH to function in transcription initiation, while only the core complex functions in DNA repair. The seven-subunit core contains Xeroderma pigmentosum group B (XPB) protein, Xeroderma pigmentosum D (XPD) protein, p62, p52, p44, p34, and p8. The CAK sub-complex includes the CDK7, Cyclin H, and MAT1 subunits. Three subunits of TFIIH are associated with enzymatic activities: SF2-family DNA-dependent ATPase/helicase activities residing in XPB and XPD, and cyclin-dependent protein kinase activity displayed by CDK7 (reviewed in [15-16]). While the enzymatic function of XPD is dedicated solely to DNA repair [17], XPB activity is required to help promoter opening during transcription initiation [18-20]. It is thought that upon ATP hydrolysis, XPB undergoes a large conformational change that has been implicated in the stable anchoring to DNA [21-22]. It appears that XPB functions in NER as a double-stranded DNA translocase that tracks along one of the two DNA strands in the 5′–3′ direction [20], leading to unwinding of the DNA duplex. The resulting single-stranded DNA (ssDNA) segment then serves for the subsequent binding by XPD, which may further extend the unwinding and scan a DNA strand to verify the presence of lesions.

TFIIH interacts with XPC-HR23B and loads on DNA near the lesion via its XPB subunit. Following TFIIH, XPA arrives at lesion site [6, 23], thereby completing the NER pre-incision complex assembly. XPA interacts both with TFIIH and XPC-HR23B and stabilizes the opened bubble together with the single-stranded binding protein (RPA) [6, 24]. A novel role for XPA suggested in lesion verification [25] consists in its ability to dissociate CAK from the TFIIH core, which substantially augments its helicase activity, as well as affinity for ssDNA [26]. Notably, in the presence of XPA, the helicase activity of the TFIIH core is further potentiated, and its blockage by bulky lesions is also more pronounced. It has been hypothesized that TFIIH-XPA interaction likely results in a conformational change of the TFIIH core complex and in a transition of TFIIH function from transcription to NER. However, the precise molecular basis of this is not fully understood yet. Interaction of XPA with some unusual DNA secondary structures configured within the intermediate NER complexes may play a role as well [27].

RPA activates the excision repair cross-complementation group 1 (ERCC1)-Xeroderma Pigmentosum group F (XPF) and Xeroderma Pigmentosum group G (XPG) nucleases that cleave 5′ and 3′ to the lesion, releasing a 24-32 nucleotide fragment containing the lesion [28, 29]. The former nuclease is recruited to the lesion site by XPA, while the later arrives through its interaction with TFIIH. XPG also replaces XPC in the pre-incision complex [30, 31]. The first incision 5′ to the lesion by ERCC1-XPF only takes place after the pre-incision complex assembly is completed, followed by initiation of repair synthesis, 3′ incision by XPG, completion of repair synthesis and ligation of the nick to restore the original DNA sequence. Repair synthesis is mediated by the polymerase activity of the DNA replication machinery and the new DNA fragment is sealed by DNA ligase I or IIIα‐XRCC1 (reviewed in [32]). For more comprehensive information on molecular mechanism of both NER pathways, the reader is referred to recent reviews [4, 10, 27, 32-37].

The added references:

Riedl, T.; Hanaoka, F.; Egly, J.M. The comings and goings of nucleotide excision repair factors on damaged DNA. EMBO J. 2003, 22, 5293-5303.

Tapias, A.; Auriol, J.; Forget, D.; Enzlin, J.H.; Schärer, O.D.; Coin, F., et al. Ordered conformational changes in damaged DNA induced by nucleotide excision repair factors. J. Biol. Chem. 2004, 279, 19074-19083.

Min, J.H.; Pavletich, N.P. Recognition of DNA damage by the Rad4 nucleotide excision repair protein. Nature 2007, 449, 570-575.

Chen, X.; Velmurugu, Y.; Zheng, G.; Park, B.; Shim, Y.; Kim, Y., et al. Kinetic gating mechanism of DNA damage recognition by Rad4/XPC. Nat. Commun. 2015, 6, 5849.

Mu, H.; Geacintov, N.E.; Broyde, S.; Yeo, J.-E.; Schärer, O.D. Molecular basis for damage recognition and verification by XPC-RAD23B and TFIIH in nucleotide excision repair. DNA Repair (Amst.) 2018, 71, 33-42.

Chu, G.; Chang, E. Xeroderma pigmentosum group E cells lack a nuclear factor that binds to damaged DNA. Science 1988, 242, 564-567.

Keeney, S.; Chang, G.J.; Linn, S. Characterization of a human DNA damage binding protein implicated in xeroderma pigmentosum E. J. Biol. Chem. 1993, 268, 21293-21300.

Wittschieben, B.Ø.; Iwai, S.; Wood, R.D. DDB1-DDB2 (xeroderma pigmentosum group E) protein complex recognizes a cyclobutane pyrimidine dimer, mismatches, apurinic/apyrimidinic sites, and compound lesions in DNA. J. Biol. Chem. 2005, 280, 39982-39989.

Moser, J.; Volker, M.; Kool, H.; Alekseev, S.; Vrieling, H.; Yasui, A., et al. The UV-damaged DNA binding protein mediates efficient targeting of the nucleotide excision repair complex to UV-induced photo lesions. DNA Repair (Amst.) 2005, 4, 571-582.

Compe, E.; Egly, J.M. Nucleotide excision repair and transcriptional regulation: TFIIH and beyond. Annu. Rev. Biochem. 2016, 85, 265-290.

Greber, B.J.; Nogales, E. The Structures of eukaryotic transcription pre-initiation complexes and their functional implications. Subcell. Biochem. 2019, 93, 143-192.

Kuper, J.; Braun, C.; Elias, A.; Michels, G.; Sauer, F.; Schmitt, D.R., et al. In TFIIH, XPD helicase is exclusively devoted to DNA repair. PLoS Biol. 2014, 12, e1001954.

Kim, T.K.; Ebright, R.H.; Reinberg, D. Mechanism of ATP-dependent promoter melting by transcription factor IIH. Science 2000, 288, 1418-1422.

Grünberg, S.; Warfield, L.; Hahn, S. Architecture of the RNA polymerase II preinitiation complex and mechanism of ATP-dependent promoter opening. Nat. Struct. Mol. Biol. 2012, 19, 788-796.

Fishburn, J.; Tomko, E.; Galburt, E.; Hahn, S. Double-stranded DNA translocase activity of transcription factor TFIIH and the mechanism of RNA polymerase II open complex formation. Proc. Natl. Acad. Sci. USA 2015, 112, 3961-3966.

Fan, L.; Arvai, A.S.; Cooper, P.K.; Iwai, S.; Hanaoka, F.; Tainer, J.A. Conserved XPB core structure and motifs for DNA unwinding: implications for pathway selection of transcription or excision repair. Mol. Cell 2006, 22, 27-37.

Egly, J.M.; Coin, F. A history of TFIIH: two decades of molecular biology on a pivotal transcription/repair factor. DNA Repair (Amst.) 2011, 10, 714-721.

Volker, M.; Moné, M.J.; Karmakar, P.; van Hoffen, A.; Schul, W.; Vermeulen, W., et al. Sequential assembly of the nucleotide excision repair factors in vivo. Mol. Cell 2001, 8, 213-224.

Li, C.-L.; Golebiowski, F.M., Onishi, Y.; Samara, N.L.; Sugasawa, K.; Yang, W. Tripartite DNA lesion recognition and verification by XPC, TFIIH, and XPA in nucleotide excision repair. Mol. Cell 2015, 59, 1025-1034.

Coin, F.; Oksenych, V.; Mocquet, V.; Groh, S.; Blattner, C.; Egly, J.M. Nucleotide excision repair driven by the dissociation of CAK from TFIIH. Mol. Cell 2008, 31, 9-20.

Sugasawa, K. Mechanism and regulation of DNA damage recognition in mammalian nucleotide excision repair. In DNA Repair; Zhao, L., Kaguni, L. S., Eds.; Elsevier Inc.: Cambridge, MA, USA, 2019; pp. 99-138.

Araujo, S.J.; Nigg, E.A.; Wood, R.D. Strong functional interactions of TFIIH with XPC and XPG in human DNA nucleotide excision repair, without a preassembled repairosome. Mol. Cell. Biol. 2001, 21, 2281-2291.

Ito, S.; Kuraoka, I.; Chymkowitch, P.; Compe, E.; Takedachi, A.; Ishigami, C., et al. XPG stabilizes TFIIH, allowing transactivation of nuclear receptors: Implications for Cockayne syndrome in XP-G/CS patients. Mol. Cell 2007, 26, 231-243.

  1. The section reviewing XPA polymorphisms and cancer incidence and treatment outcome can be presented without Tab.1, and the length of this section can be shortened by displaying the most significant part of it.

We insist on keeping Table 1 because some data, that Table 1 contains, are not presented in the text. Originally, text on XPA polymorphisms and cancer incidence and treatment outcome was much longer and we already shortened it prior to first submission. Therefore, the present extent of this text seems to us optimal.

  1. Fig.1 clearly shows how XPA expression is regulated by different physiological conditions, but A and 1B should be reversed according to the text.

Corrected, as suggested.

  1. How XPA functions as a scaffold protein in NER should be expanded in the beginning of section 2 and sentences regarding interacting amino acid residues in XPA or other factors should be kept to a minimum to avoid boring the readers.

Based on your suggestion, sentences regarding interacting amino acid residues in XPA or other factors have been shortened in some cases. Additional information on scaffold function of XPA has been added to our manuscript. The related text reads:

XPA functions in conjunction with RPA as the scaffold for the assembly and stabilization of the NER pre-incision complex, organizing the damaged DNA and this complex to ensure lesions are appropriately excised. Interaction of the two proteins is mediated by the 32 kDa subunit of RPA (RPA32) and a motif in the disordered N-terminal domain of XPA [52]. Because RPA32 is tethered to the DNA substrate binding apparatus by a 33 aa flexible linker [53] and RPA32-binding motif on XPA resides in the flexible, disordered N-terminal domain, about 50 residues from MBD [52, 54], it is still mysterious how activities of these two proteins are coordinated by their interaction. It has been proposed that the critical factor enabling the coordination of XPA and RPA is the direct physical interaction between MBD of XPA and 70 kDa subunit of RPA (RPA70), as this interaction is spatially proximate to the binding of both proteins to the NER bubble [55].

The added references:

Mer, G.; Bochkarev, A.; Gupta, R.; Bochkareva, E.; Frappier, L.; Ingles, C.J.; Edwards, A.M.; Chazin, W.J. Structural basis for the recognition of DNA repair proteins UNG2, XPA, and RAD52 by replication factor RPA. Cell 2000, 103, 449-456.

Brosey, C.A.; Chagot, M.E.; Ehrhardt, M.; Pretto, D.I.; Weiner, B.E.; Chazin, W.J. NMR analysis of the architecture and functional remodeling of a modular multidomain protein, RPA. J. Am. Chem. Soc. 2009, 131, 6346-6347.

Ali, S.I.; Shin, J.S.; Bae, S.H.; Kim, B.; Choi, B.S. Replication protein A 32 interacts through a similar binding interface with TIPIN, XPA, and UNG2. Int. J. Biochem. Cell. Biol. 2010, 42, 1210-1215.

Topolska-Woś, A.M.; Sugitani, N.; Cordoba, J.J.; Le Meur, K.V.; Le Meur, R.A.; Kim, H.S.; Yeo, J.-E.; Rosenberg, D.; Hammel, M.; Schärer, O.D., Chazin, W.J. A key interaction with RPA orients XPA in NER complexes. Nucleic Acids Res. 2020, 48, 2173-2188.

Reviewer 2 Report

XPA: DNA Repair Protein of Significant Clinical Importance

This review paper highlights the interacting proteins and regulatory mechanisms of XPA, a key factor for the NER mechanism. It is well established that loss of NER function is linked to tumorigenesis as well as inhibition of NER activity during chemotherapy, especially with cisplatin, which has been producing a favorable outcome in certain types of cancer. Accordingly, it is not difficult to find review papers on the details of the NER mechanism and its application in cancer therapy as well. This paper is also touching some of the overlapping topics mentioned above. However, this paper instead focused specifically on the single NER factor, XPA. Both canonical and non-canonical functions of XPA is relatively well discussed in this paper. In general, this is a well-prepared manuscript and suitable for the readers of the journal. However, before I recommend this article to be published in IJMS, the authors should address the following points:

  1. The authors should add the following references and discuss how these data can be reconciled with XPA function and expression in tumorigenesis and/or cancer therapy.
    (1) Some of XPA binding partners having roles outside NER were introduced in Section 5. (2) In Section 6. A circadian protein cryptochrome has been shown to regulate cisplatin repair kinetics as well (Kang, 2014, Nucleic Acids Res).
  2. (3) In Section 7. The functional consequence of between the increased SIRT1 and the level of deacetylated XPA has been shown in a recent publication (Choi, 2015, Oncotarget).
  3. NDR1 protein belongs to this category that needs to be discussed (Park, 2015, BBRC),
  4. Grammatical errors throughout the manuscript, such as different font sizes, misuse of underlines, and subscript et al., should be corrected.

Author Response

This review paper highlights the interacting proteins and regulatory mechanisms of XPA, a key factor for the NER mechanism. It is well established that loss of NER function is linked to tumorigenesis as well as inhibition of NER activity during chemotherapy, especially with cisplatin, which has been producing a favorable outcome in certain types of cancer. Accordingly, it is not difficult to find review papers on the details of the NER mechanism and its application in cancer therapy as well. This paper is also touching some of the overlapping topics mentioned above. However, this paper instead focused specifically on the single NER factor, XPA. Both canonical and non-canonical functions of XPA is relatively well discussed in this paper. In general, this is a well-prepared manuscript and suitable for the readers of the journal. However, before I recommend this article to be published in IJMS, the authors should address the following points:

  1. The authors should add the following references and discuss how these data can be reconciled with XPA function and expression in tumorigenesis and/or cancer therapy.
    (1) Some of XPA binding partners having roles outside NER were introduced in Section 5. (2) In Section 6. A circadian protein cryptochrome has been shown to regulate cisplatin repair kinetics as well (Kang, 2014, Nucleic Acids Res).
  2. (3) In Section 7. The functional consequence of between the increased SIRT1 and the level of deacetylated XPA has been shown in a recent publication (Choi, 2015, Oncotarget).
  3. NDR1 protein belongs to this category that needs to be discussed (Park, 2015, BBRC).

All suggested references have been included and the added data have been reconciled with XPA function and expression in tumorigenesis and/or cancer therapy, as proposed.

  1. With respect to circadian time-dependent removal of CDDP-induced DNA damage, the following sentence and reference has been added to our manuscript:

In addition, in mouse liver cells, the removal of CDDP-induced DNA damage is strongly dependent on circadian time [110].

Kang, T.-H.; Leem, S.-H. Modulation of ATR-mediated DNA damage checkpoint response by cryptochrome 1. Nucleic Acids Res. 2014, 42, 4427-4434.

  1. Regarding SIRT1 and the level of deacetylated XPA, we have added the following text and references:

Accordingly, it has been shown that increased levels of SIRT1 result in an increase in deacetylated XPA that persists in chromatin-bound state and facilitates repair of UV- and CDDP-induced DNA lesions [135]. Expectedly, increased levels of SIRT1 in cancer cells can confer CDDP resistance and thus represent treatment obstacle [133, 135] that, however, can be overcome by targeting by SIRT1 inhibitors [136-138].

Choi, J.Y.; Park, J.-M.; Yi, J.M.; Leem, S.-H.; Kang, T.-H. Enhanced nucleotide excision repair capacity in lung cancer cells by preconditioning with DNA-damaging agents. Oncotarget 2015, 6, 22575-22586.

Brunet, A.; Sweeney, L.B.; Sturgill, J.F.; Chua, K.F.; Greer, P.L., et al. Stress-dependent regulation of FOXO transcription factors by the SIRT1 deacetylase. Science 2004, 303, 2011-2015.

Zhao, W.; Kruse, J.P.; Tang, Y.; Jung, S.Y.; Qin, J., et al. Negative regulation of the deacetylase SIRT1 by DBC1. Nature 2008, 451, 587-590.

Lin, Z.; Yang, H.; Kong, Q.; Li, J.; Lee, S.M., et al. USP22 antagonizes p53 transcriptional activation by deubiquitinating Sirt1 to suppress cell apoptosis and is required for mouse embryonic development. Mol. Cell 2012, 46, 484-494.

  1. In case of NDR1, we have added the following text and references:

Members of the nuclear-Dbf2-related (NDR) family of serine/threonine kinases are highly conserved from yeasts to humans and function in processes associated with cell cycle regulation, including centrosome duplication, apoptosis, and the alignment of mitotic chromosomes [89]. Although XPA interacts with NDR1, biological relevance of this interaction is largely unknown yet. It appears that NDR1 regulates the removal of XPA from the chromatin without directly associating with the chromatin. Mechanically, this could be achieved by its regulation of the ATR-mediated DDR pathway [90].

Hergovich, A.; Cornils, H.; Hemmings, B.A. Mammalian NDR protein kinases: from regulation to a role in centrosome duplication. Biochim. Biophys. Acta 2008, 1784, 3e15.

Park, J.-M.; Choi, J.Y.; Yi, J.M.; Chung, J.W.; Leem, S.-H., et al. NDR1 modulates the UV-induced DNA-damage checkpoint and nucleotide excision repair. Biochem. Biophys. Res. Com. 2015, 461, 543-548.

  1. Grammatical errors throughout the manuscript, such as different font sizes, misuse of underlines, and subscript et al., should be corrected.

We did our best to remove all grammatical errors throughout the manuscript. Please, check the revised version of our manuscript to see the changes.

Reviewer 3 Report

The manuscript is a detailed review on XPA protein, focusing not only on its role in NER, but also on its interaction partners, regulations, and implications in cancer development and therapy. This approach is rather original and similar recent review has not been published. The review covers the topic in great detail, which is commendable. It is my opinion that information-wise the manuscript does not require major changes and describes current state-of art. However, as this also includes a lot of descriptive information which is hard to go through for lot of readers, I would suggest

  • Including a short summary at the end of each section, to summarize the findings and implicate consequences
  • Including graphical scheme to accompany sections on interacting partners (3 and 5)
  • Modifying scheme in Figure 1 to fit on one page; the arrangement of panels side-by-side would allow better comparison among the three highlighted regulation pathways
  • Include a scheme on posttranslational regulations (section 7)
  • Consider including structural formulas of mentioned inhibitors (section 8)

Author Response

The manuscript is a detailed review on XPA protein, focusing not only on its role in NER, but also on its interaction partners, regulations, and implications in cancer development and therapy. This approach is rather original and similar recent review has not been published. The review covers the topic in great detail, which is commendable. It is my opinion that information-wise the manuscript does not require major changes and describes current state-of art. However, as this also includes a lot of descriptive information which is hard to go through for lot of readers, I would suggest

  1. Including a short summary at the end of each section, to summarize the findings and implicate consequences.

As no other reviewer asked this type of change of our manuscript, we rather avoided of this. Nevertheless, there are sections which do contain a short summary at the end. In addition, there are some sections for which it is nearly impossible to bring a short summary. Thanks a lot for your understanding.

  1. Including graphical scheme to accompany sections on interacting partners (3 and 5).

Done, as requested.

  1. Modifying scheme in Figure 1 to fit on one page; the arrangement of panels side-by-side would allow better comparison among the three highlighted regulation pathways.

Done, as requested.

  1. Include a scheme on posttranslational regulations (section 7).

Done, as requested.

  1. Consider including structural formulas of mentioned inhibitors (section 8).

We better try to avoid this, as it may require permission both from authors and journals that firstly published these formulas. Nevertheless, to meet your suggestion, we have included sentence into our manuscript clearly referring the reader to original papers. This sentence reads:

For detailed information on structural formulas of all mentioned XPA inhibitors, the reader is referred to original papers [148-150, 152, 153].

Reviewer 4 Report

Pulzová et al have presented a comprehensive review of XPA's role in DNA repair and its clinical significance. The manuscript is organized very well and very well written. However, a major deficiency is the lack of any figures describing XPA protein or NER pathway. I request the authors to include a figure depicting the role of XPA in NER and its known protein interactors. This will improve this manuscript.

Author Response

Pulzová et al have presented a comprehensive review of XPA's role in DNA repair and its clinical significance. The manuscript is organized very well and very well written. However, a major deficiency is the lack of any figures describing XPA protein or NER pathway. I request the authors to include a figure depicting the role of XPA in NER and its known protein interactors. This will improve this manuscript.

Based on your and other reviewers suggestion, we have added two figures to the manuscript. First of the added figures shows a domain structure of XPA, as well as domains responsible for its interaction with other proteins. The second depicts post-translational modifications of this protein. We decided not to include figure showing the role of XPA in NER, because this can be found in several very recent and comprehensive reviews, as stated in the manuscript. As noticed, mechanistic details of NER are a bit out of scope of our paper. Rather, association of XPA with cancer incidence and treatment outcome is the main focus.

Round 2

Reviewer 1 Report

ijms 726807 reviewing the function of XPA in nucleotide excision repair and its significance as a cancer prognostic biomarker has been carefully revised according to the comments of reviewers. Many new referrences have been incorporated into the revised version to strengthen the description of NER-linked DNA damage recognition and the whole manuscript now reads quite well. The revised paper can be accepted for publication in IJMS after a final check of language. 

Author Response

As recommended, native English-speaking co-author Thomas A. Ward did a final language check. Please, see our second revision and track changes.